# COVID-19 vaccination in Sindh Province, Pakistan: A modelling study of health impact and cost-effectiveness

Carl A. B. Pearson[1,2]*, Fiammetta Bozzani[3], Simon R. Procter[1,3], Nicholas G. Davies[1], Maryam Huda[4], Henning Tarp Jensen[3], Marcus Keogh-Brown[3], Muhammad Khalid[5], Sedona Sweeney[3], Sergio Torres-Rueda[3], CHiL COVID-19 Working Group[3¶], CMMID COVID-19 Working Group[1¶], Rosalind M. Eggo[1], Anna Vassall[3], Mark Jit[1,3]

1 Centre for Mathematical Modelling of Infectious Diseases, London School of Hygiene and Tropical Medicine, London, United Kingdom, 2 South African DSI-NRF Centre of Excellence in Epidemiological Modelling and Analysis (SACEMA), Stellenbosch University, Stellenbosch, Republic of South Africa, 3 Centre for Health Economics in London, London School of Hygiene and Tropical Medicine, London, United Kingdom, 4 Aga Khan University Hospital, Karachi, Sindh, Pakistan, 5 Ministry of National Health Services Regulations & Coordination Islamabad, Pakistan

☯ These authors contributed equally to this work.
¶ Membership of CHiL COVID-19 Working Group and CMMID COVID-19 Working Group is provided in the Acknowledgements.
* carl.pearson@lshtm.ac.uk

**Data Availability Statement:** The code and input data are available from https://github.com/cmmid/vaxco. Simulation outputs are available from https://doi.org/10.5281/zenodo.5070957.

## Abstract

### Background

Multiple Coronavirus Disease 2019 (COVID-19) vaccines appear to be safe and efficacious, but only high-income countries have the resources to procure sufficient vaccine doses for most of their eligible populations. The World Health Organization has published guidelines for vaccine prioritisation, but most vaccine impact projections have focused on high-income countries, and few incorporate economic considerations. To address this evidence gap, we projected the health and economic impact of different vaccination scenarios in Sindh Province, Pakistan (population: 48 million).

### Methods and findings

We fitted a compartmental transmission model to COVID-19 cases and deaths in Sindh from 30 April to 15 September 2020. We then projected cases, deaths, and hospitalisation outcomes over 10 years under different vaccine scenarios. Finally, we combined these projections with a detailed economic model to estimate incremental costs (from healthcare and partial societal perspectives), disability-adjusted life years (DALYs), and incremental cost-effectiveness ratio (ICER) for each scenario.

We project that 1 year of vaccine distribution, at delivery rates consistent with COVAX projections, using an infection-blocking vaccine at $3/dose with 70% efficacy and 2.5-year duration of protection is likely to avert around 0.9 (95% credible interval (CrI): 0.9, 1.0) million cases, 10.1 (95% CrI: 10.1, 10.3) thousand deaths, and 70.1 (95% CrI: 69.9, 70.6)

**Funding:** The following funding sources are acknowledged as providing funding for the named authors. This research was partly funded by the Bill & Melinda Gates Foundation (INV-003174: MJ; INV-016832: SRP; NTD Modelling Consortium OPP1184344: CABP). This project has received funding from the European Union's Horizon 2020 research and innovation programme - project EpiPose (101003688: MJ). FCDO/Wellcome Trust (Epidemic Preparedness Coronavirus research programme 221303/Z/20/Z: CABP). HDR UK (MR/S003975/1: RME). This research was partly funded by the National Institute for Health Research (NIHR) using UK aid from the UK Government to support global health research. The views expressed in this publication are those of the author(s) and not necessarily those of the NIHR or the UK Department of Health and Social Care (16/137/109: MJ; NIHR200908: RME; NIHR200929: MJ, NGD). UK MRC (MC_PC_19065 - Covid 19: Understanding the dynamics and drivers of the COVID-19 epidemic using real-time outbreak analytics: NGD, RME). UKRI Research England (NGD). The funders had no role in study design, data collection and analysis, decision to publish, or preparation of the manuscript.

**Competing interests:** The authors have declared that no competing interests exist.

**Abbreviations:** CEPI, Coalition for Epidemic Preparedness Innovations; COVID-19, Coronavirus Disease 2019; CrI, credible interval; DALY, disability-adjusted life year; DIC, deviance information criterion; EPI, Expanded Program on Immunization; GDP, gross domestic product; ICER, incremental cost-effectiveness ratio; LMIC, low- and middle-income country; NPI, nonpharmaceutical intervention; QALY, quality-adjusted life year; SAGE, Strategic Group of Experts on Immunization; SARS-CoV-2, Severe Acute Respiratory Syndrome Coronavirus 2; WHO, World Health Organization; YLL, year of life lost.

thousand DALYs, with an ICER of $27.9 per DALY averted from the health system perspective. Under a broad range of alternative scenarios, we find that initially prioritising the older (65+) population generally prevents more deaths. However, unprioritised distribution has almost the same cost-effectiveness when considering all outcomes, and both prioritised and unprioritised programmes can be cost-effective for low per-dose costs. High vaccine prices ($10/dose), however, may not be cost-effective, depending on the specifics of vaccine performance, distribution programme, and future pandemic trends.

The principal drivers of the health outcomes are the fitted values for the overall transmission scaling parameter and disease natural history parameters from other studies, particularly age-specific probabilities of infection and symptomatic disease, as well as social contact rates. Other parameters are investigated in sensitivity analyses.

This study is limited by model approximations, available data, and future uncertainty. Because the model is a single-population compartmental model, detailed impacts of non-pharmaceutical interventions (NPIs) such as household isolation cannot be practically represented or evaluated in combination with vaccine programmes. Similarly, the model cannot consider prioritising groups like healthcare or other essential workers. The model is only fitted to the reported case and death data, which are incomplete and not disaggregated by, e.g., age. Finally, because the future impact and implementation cost of NPIs are uncertain, how these would interact with vaccination remains an open question.

## Conclusions

COVID-19 vaccination can have a considerable health impact and is likely to be cost-effective if more optimistic vaccine scenarios apply. Preventing severe disease is an important contributor to this impact. However, the advantage of prioritising older, high-risk populations is smaller in generally younger populations. This reduction is especially true in populations with more past transmission, and if the vaccine is likely to further impede transmission rather than just disease. Those conditions are typical of many low- and middle-income countries.

## Author summary

### Why was this study done?

- The evidence base for health and economic impact of Coronavirus Disease 2019 (COVID-19) vaccination in low- and middle-income settings is limited.

- Searching PubMed, medRxiv, and econLit using the search term ("coronavirus" OR "covid" OR "ncov") AND ("vaccination" OR "immunisation") AND ("model" OR "cost" OR "economic") for full text articles published in any language between 1 January 2020 and 20 January 2021, returned 29 (PubMed), 1,167 (medRxiv), and 0 (econLit) studies: 20 overall were relevant, with only 4 exclusively focused on low- or middle-income countries (India, China, Mexico), while 3 multicountry analyses also included low- or middle-income settings.

- However, only 3 of these studies are considered economic outcomes, all of them comparing the costs of vaccination to the costs of nonpharmaceutical interventions (NPIs)

and concluding that both are necessary to reduce infections and maximise economic benefit.

- The majority of studies focus on high-income settings and conclude that prioritizing COVID-19 vaccination to older age groups is the preferred strategy to minimise mortality, particularly when vaccine supplies are constrained, while other age- or occupational risk groups should be prioritised when vaccine availability increases or when other policy objectives are pursued.

## What did the researchers do and find?

- We combined epidemiological and economic analysis of COVID-19 vaccination based on real-world disease and programmatic information in the Sindh Province of Pakistan.

- We found that vaccination in this setting is likely to be highly cost-effective, and even cost saving, as long as the vaccine is reasonably priced and efficacy is high.

- Unlike studies in high-income settings, we also found that vaccination programmes targeting all adults may have almost as much benefit as those initially targeted at older populations, likely reflecting the higher previous infection rates and different demography in these settings.

## What do these findings mean?

- The results suggest that low- and middle-income countries (LMICs) see less benefit to initially prioritising vaccination of older (65+) populations compared to unprioritised distribution. Factors outside this analysis, like cost differences between prioritised and unprioritised programmes, will further influence the preferred approach.

- As such, LMICs and international bodies providing guidance for LMICs need to consider evidence specific to these settings when making recommendations about COVID-19 vaccination.

- Further data and model-based analyses in such settings are urgently needed in order to ensure that vaccination decisions are appropriate to these contexts.

## Introduction

The Coronavirus Disease 2019 (COVID-19) pandemic has resulted in over 50 million cases and nearly 2 million deaths in 2020, with cases in nearly every country [1]. To reduce transmission of the causal Severe Acute Respiratory Syndrome Coronavirus 2 (SARS-CoV-2) virus, many countries have imposed physical distancing measures such as closure of schools and workplaces and restrictions on public gatherings [2]. Such measures often incur socioeconomic costs that are not indefinitely sustainable, particularly in resource-poor settings [3], and, when these measures are lifted, transmission has readily resumed in most places [4].

Vaccination may provide a durable option to protect individuals. If a vaccine also reduces transmission (e.g., by preventing infection or limiting infectiousness of disease), even unvaccinated individuals would have reduced infection risk. As of January 2021, 3 vaccines have

completed Phase III trials, and at least 20 other vaccine candidates were in Phase III trials, with over 250 in earlier trials or preclinical studies [5,6].

Many high-income and large middle-income countries have signed bilateral agreements with manufacturers, preordering enough vaccines to cover their populations, some multiple times [7]. The World Health Organization (WHO), Gavi, the Vaccine Alliance, and the Coalition for Epidemic Preparedness Innovations (CEPI) launched the COVAX Facility to enable many small and lower-income countries to pool their purchasing power. To date, 141 countries and territories have started the COVAX participation process, which will distribute vaccines to participating countries according to population size [8,9]. This distribution will cover a small proportion (3%) of those populations in the months following vaccine approval, aiming to expand to 20% by the end of 2021 [10]. Given that this supply may not be sufficient, countries may need to purchase additional vaccines, and to do so need to be able to quantify the costs and benefits of various vaccination programmes to compare to other health sector investments. Additionally, countries face other substantial health sector resource constraints that may require external funding including scale-up of vaccine delivery infrastructure and workforce and continued care and treatment of those with COVID-19.

WHO's Strategic Group of Experts on Immunization (SAGE) has issued a roadmap to help countries prioritise distribution of these limited doses. This roadmap draws from work across multiple disciplines, including modelling to project the health outcomes, health sector financing, and broader economic consequences of different vaccine prioritisation strategies, but much of the available research has focused on high-income settings [11–14].

To address this gap, we assessed the health impact, economic impact, and cost-effectiveness of COVID-19 vaccination in Sindh Province, Pakistan, using a combined epidemiological and economic model. We chose a specific setting to ensure that our model could incorporate local mobility and cost data. Sindh Province initially confirmed a large number of cases, followed by declining incidence after a nationwide lockdown. Our analysis addresses vaccine prioritisation questions faced by both global (WHO) and national (Pakistan Ministry of Health) decision-makers and illustrates the decision support analysis that should be applied more broadly in low- and middle-income countries (LMICs).

## Methods

### Ethics

All clinical data used were obtained from publicly available sources, so no ethical approval was required for this study.

### Epidemiological model

To capture the natural history and transmission of SARS-CoV-2, we used a previously published compartmental model [15–17] tailored to the population of Sindh using population data from WorldPop [18] and assumed baseline population contact rates from previously estimated national patterns for Pakistan [19].

Briefly, the model compartments are an extended SEIRS+V (*S*usceptible, *E*xposed, *I*nfectious with multiple subcompartments, *R*ecovered and/or *V*accinated, potentially converting either to *S*usceptible or if in the combined state to only *R*ecovered or *V*accinated) system with births, deaths, and age structure. For all compartments other than *R*ecovered and/or *V*accinated, we use event-time distributions derived from global observations (Table 1). For *R*ecovered and/or *V*accinated, we consider multiple characteristic protection durations, given the uncertainty in these durations. For the *R*ecovered compartment (with and without vaccination), we assume perfect protection; we address the *V*accinated compartment protection along with the vaccination programme details in the "Vaccine programme" section. The modelled

**Table 1. Summary of epidemiological, vaccine, and economic parameters used in the base case and scenario analysis.**

| Parameter | Base case value | Scenario range | Source |
|---|---|---|---|
| *Epidemiological parameters* | | | |
| Latent period | Gamma(mean = 2.5, k = 5) | | [41,42] |
| Contact rates | Age-dependent synthetic contact matrix for Pakistan | | [19] |
| Proportion asymptomatic | age-specific | | posterior from [15] |
| Duration of infectiousness | Gamma(mean = 5, k = 4) | | [41,42] |
| Duration of natural immunity | 2.5 years | 1 year, lifelong | Assumed |
| *Vaccine-related parameters* | | | |
| Duration of vaccine programme | 1 year | 5 and 10 years | |
| Initial age targeting | 65+ | 15+ | |
| Duration of vaccine-induced immunity | 2.5 years | 1 year, lifelong | Assumed |
| Initial number of courses administered per day | 4,000 (based on COVAX availability; see S1 Text) | 8,000, 12,000, 184,000 (short campaign only) | [25] |
| Number of doses per course | 2 | 1 | [25] |
| Efficacy | 70% | 30%, 90% | Assumed |
| *Economic parameters* | | | |
| Vaccine procurement price per dose | $3 | $6 and $10 | [8] |
| Wastage | 15% of vaccine procurement price per dose, 10% of immunisation supplies procurement price per dose | | [36]; see S1 Text for details |
| Freight | 10% of vaccine procurement price per dose | | Assumed |
| Syringes and safety boxes | 0.04 | | UNICEF supply division price list |
| Cold chain costs per dose (national level) | $0.133 | | [37] |
| Cold chain costs per dose (service level) | $0.029 | | Microcosting. Various sources (see S1 Text). |
| Human resource per dose | $0.38 | | Microcosting from DCP project [38,39] |
| Transport per dose | Transport to facility: $0.04 Transport from facility to campaign site: $0.001 | | Microcosting. Various sources (see S1 Text). |
| Social mobilisation per dose | $0.16 | | [40] |
| Health system markup | 31% of cost per dose excluding procurement price per dose, immunisation supplies, wastage, and freight | | [37] |
| Perspective | Health system | Societal | |
| Discount rate | 3% costs, 3% DALYs | 3% costs, 0% DALYs | [35] |

US dollars abbreviated as $ and disability-adjusted life years as DALYs.

loss of protection can represent a range of phenomena, from antibody waning to shifts in the circulating pathogen with time leading to immune escape.

We assumed that contact patterns changed over the course of the epidemic, and estimated these changes using Google Community Mobility indicators [20] for Sindh and school closures as reflected in the Oxford Coronavirus Government Response Tracker [2]. For projections, we assume that contact patterns return to the baseline contract matrix at the end of May 2021, and no further physical distancing interventions are imposed.

## Model fitting and projections

Using Bayesian inference via Markov Chain Monte Carlo, we fit 5 elements of the model: the effective introduction date, $t_0$, as number of days after 1 January 2020; the basic reproduction

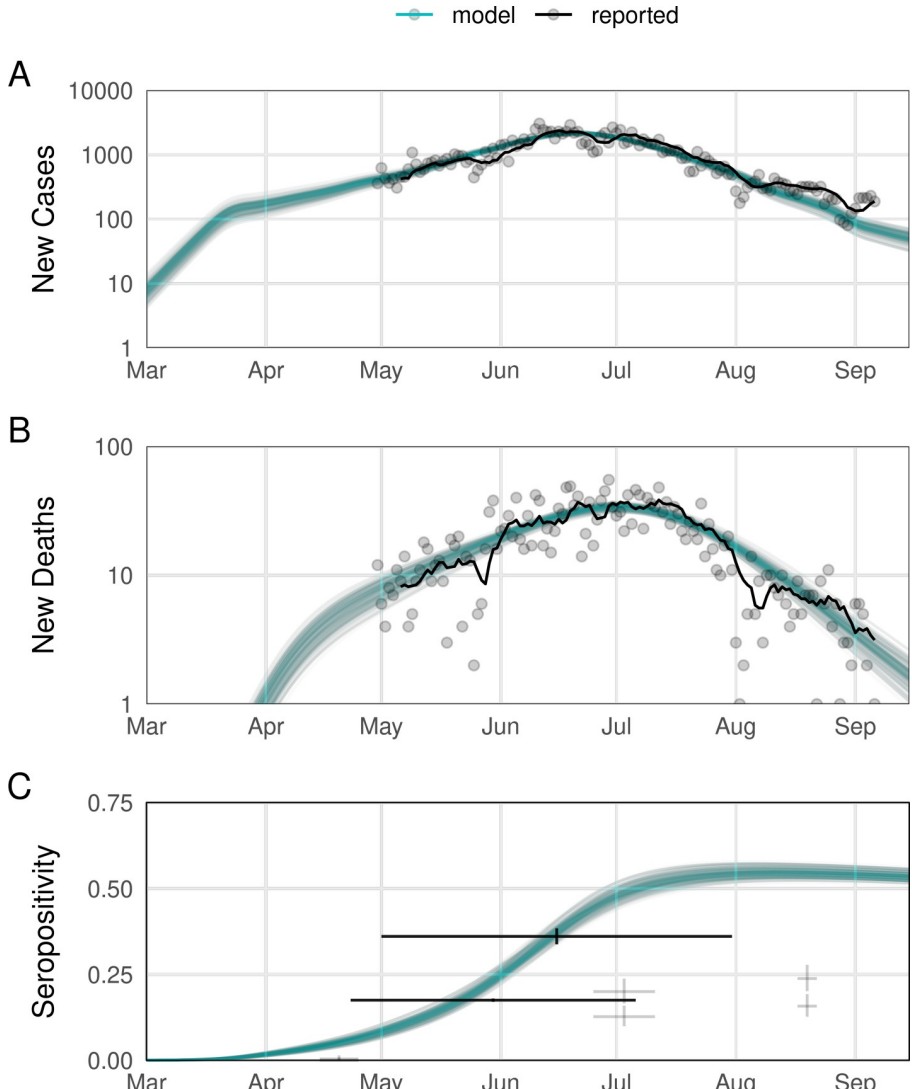

**Fig 1. Outcomes for fitted model ascertained outcomes compared to data.** Sample ascertained trajectories ($n = 250$) from the posterior of model parameters (blue) compared to observed outcomes (black). For observed cases and deaths, the solid line is the 7-day average, with points corresponding to daily reports. For the limited serological data, the crosshairs show the collection period and binomial confidence interval on the seropositivity estimates. The serial study results with expected low seropositivity are faded. Expected duration of infection-derived immunity assumed to be 2.5 years; other immunity assumptions in Fig C in S1 Text. All of the assumptions considered produce comparable fits to reported cases and deaths through September 2020.

number in Sindh without any interventions, $R_0$; a time-varying ascertainment rate for both COVID-19 deaths and cases; and the standard deviation characterising the distribution of reported data points around the model-predicted mean value. We fitted the model to the new daily cases and deaths in Sindh reported by the Government of Pakistan COVID-19 Dash-board [21] from 30 April to 14 September 2020 (Fig 1A and 1B).

As a validation, we compared model outputs to 3 reports of SARS-CoV-2 seroprevalence (Fig 1C), all from Karachi [22–24]. Two concerned broad population samples over an extended period [23,24]: These generally overlap with model estimates but are aggregated over time making precise comparison difficult. The third conducted repeat surveys in 2 specific

regions [22], finding clear qualitative trends that match model trends—limited exposure pre-May, rapid rise through July, and subsequent plateau—but recorded lower levels. Additional figures in that study, however, indicate that the specific study sites recorded lower positivity trends generally than their broader districts, and particularly during the case surge in June, suggesting that the measured values may be lower than those in the broader population. As another out-of-sample validation, we also compared forward projections to Sindh data for 15 September 2020 to 15 January 2021.

Since the waning rate of infection-acquired immunity is unknown and the benefits of vaccination highly sensitive to this parameter, we repeated the fitting exercise with 4 assumptions for waning of infection-acquired immunity: lifelong and exponentially waning immunity with expected durations of 1, 2.5, and 5 years.

### Vaccine programme

We assumed vaccination distribution consistent with the availability of doses indicated by WHO SAGE's Working Group on COVID-19 Vaccines [25]. We assumed that vaccination required 2 doses per course, since this is true of most vaccines in the COVAX portfolio.

The number of full courses available each month was assumed to be divided among all COVAX participating countries proportional to population, and likewise for subnational regions. Additionally, we assumed that 15% of courses would be wasted for reasons such as cold chain failures, incorrect use, or failure to complete second doses (which we pessimistically assume means lack of vaccine protection). Hence, we assumed that Sindh would complete 4,000 courses/day in the first 3 months (with a 30-day delay accounting for timing of second dose) after a vaccine is approved. We assumed that courses delivered would increase to 16,000, 24,000, then 32,000 in subsequent quarters using the schedule suggested by WHO SAGE [25] modified to reflect the current vaccine landscape (see Section E in S1 Text). For sensitivity analyses, we also consider (a) a constant 4,000 courses per day and (b) a sufficient constant rate to cover the eligible population in 6 months, 184k courses per day, which is comparable to peak rates achieved in some high-income countries (around 0.4% of the population per day).

For primary vaccine scenarios, we assumed that the vaccine is infection-blocking and that protection is complete for some individuals and absent in others (i.e., "all-or-nothing" protection); we considered other vaccine models (every exposure tests the efficacy independently, i.e., "leaky" protection, and/or disease-only blocking) as sensitivity studies. Vaccine doses are distributed among individuals in the Susceptible and Recovered compartments; Susceptible individuals become Vaccinated, and Recovered individuals become Recovered and Vaccinated.

We considered different durations of protection: Once vaccinated in the model, individuals lose vaccine protection with an exponentially distributed duration. Finally, we considered different efficacy levels of protection. A wide variety of COVID-19 vaccines are available to Pakistan via COVAX, which have reported different efficacy levels [26]. Instead of modelling a particular vaccine, for our base case scenario, we assume a vaccine with 70% efficacy that protects for 2.5 years on average. As alternatives, we considered a higher efficacy (90%) or longer duration of protection (5 years). See S1 Text for more combinations.

We track vaccine impact for 10 years and assume that vaccination continues at the same rate (after initial scale-up) for 1 (base case), 5 or 10 years, or for 0.5 year for the sensitivity study of high delivery rate. For simplicity, during the time the vaccine programme is active, vaccination occurs on each day of the week, rather than excluding weekends and holidays. We assume that vaccination cost does not fundamentally change as the programme continues and coverage increases; this implicitly assumes a linear average in the costs of vaccination across groups and time.

Given the emphasis on prioritising older adults in WHO's vaccine prioritisation roadmap [27], we considered 2 scenarios for distribution: either individuals 15+ years old for the duration or individuals 65+ years old for the first two-quarters of the first year before shifting to 15 +. For all scenarios, we assume that vaccine doses are uniformly (i.e., proportional to fraction of population) distributed in the targeted populations.

## Health and economic outcomes

We modelled the impact of COVID-19 vaccination on cases, deaths, and disability-adjusted life years (DALYs) compared to counterfactual scenarios with no vaccination over a 10-year time horizon. For different vaccination scenarios, the averted DALYs were combined with the costs of the vaccination programme and any reduction in COVID-19 case management costs from vaccination to calculate incremental cost-effectiveness ratios (ICERs). Our analysis followed the Consolidated Health Economic Evaluation Reporting Standards (CHEERS; for checklist, see Section J in S1 Text), and base case model parameters are listed in Table 1.

## DALYs

For each scenario, we modelled the health burden in DALYs for symptomatic cases, nonfatal hospitalisations, nonfatal admissions to critical care, and premature death due to COVID-19. For the nonfatal outcomes, and in the absence of specific DALY data, we used quality-adjusted life years (QALYs) reported by Sandmann and colleagues [28] based on pandemic influenza studies treated 1 QALY gained as equivalent to 1 DALY averted.

For COVID-19 deaths, we estimated DALYs, guided by the approach presented by Briggs and colleagues [29]. We generated age at death in 5-year age bands, and then applied age-specific life expectancy at death using national life tables for Pakistan (United Nations estimates for 2015 to 2020 [30]). We adjusted years of life lost (YLLs) considering the overall level of disability for any remaining years of life using data on QoL by age band from Zimbabwe [31] since all other countries with available data were high-income. However, in our base case analysis, we did not adjust standard life tables to take into account any reduced life expectancy due to specific comorbidities associated with COVID-19. As a sensitivity analysis, since risk of severe COVID-19 is higher for people with comorbidities [32], we modelled an alternative scenario in which half of COVID-19-related deaths were assumed to occur in individuals with higher baseline mortality (standardised mortality ratio = 1.5) and 10% lower baseline QoL. We calculated the average DALYs per death using both 3% (base case) and 0% discounting (Table G in S1 Text).

## Costs

We estimated annual economic costs of vaccine introduction and of diagnosis and treatment in 2020 values, using an exchange rate of 155 PKR (Pakistani Rupee) for 1 $ (US dollar) on 1 January 2020 [33] and adjusting earlier data by the gross domestic product (GDP) deflator for Pakistan [34]. Following WHO guidelines, we used a 3% discount rate for future costs and for annualising capital investments, while health outcomes are discounted at either 3% (base case) or 0% [35]. The costing was carried out from a health system (vaccination, testing, and care and treatment costs) and partial societal perspective (including household costs incurred by COVID-19 illness and case management or costs of illness, but excluding benefits of reduced nonpharmaceutical interventions (NPIs)), using a bottom-up ingredients-based approach.

**Costs of COVID-19 vaccine introduction.** It was assumed that all vaccine doses would be delivered through campaigns in the community. Vaccine and immunisation costs, including supplies costs per dose with freight charges and wastage, are in Table 1. Full costing details are given in Sections 6 and 7 in S1 Text.

The price of the COVID-19 vaccine itself was set at $3 per dose, at which the Serum Institute of India has capped prices for LMICs [36]. The cost per dose of expanding national and provincial level cold chain equipment was obtained from a model of the costs of delivering COVID-19 vaccines in the 92 COVAX countries developed by UNICEF [37]. The additional cold chain costs at the facility level were calculated by allocating a proportion of existing equipment and electricity costs to the COVID-19 vaccine relative to the volume of other vaccines in the immunisation programme.

We estimated vaccine delivery costs via a statewide campaign by adding together the costs of human resources, social mobilisation, and transport. We assumed that nurses and vaccinators would deliver the vaccines. We carried out a microcosting of human resource costs using data from the Disease Control Priorities project [38,39]. Social mobilisation costs were obtained from budgets from a poliovirus campaign [40]. Transportation costs of delivering vaccines to the distribution sites were obtained from the UNICEF model [39]. Transportation costs associated with campaigns originating at facilities were calculated by estimating the catchment areas for facilities and assuming daily vehicle journeys corresponding to the radius of the catchment area.

Our delivery costs did not include additional health system activities, such as planning and coordination, pharmacovigilance, and waste management. Accordingly, we added a 31% markup on the delivery costs, obtained from the UNICEF model [37].

**Costs of COVID-19 diagnosis and treatment.** The economic impact of COVID-19 on the health system includes diagnosis and clinical management. Costing methods and estimates are reported in full elsewhere [17,43]. Briefly, unit costs of outputs, such as bed days or outpatient visits, were sourced from a range of primary published and unpublished sources in Pakistan. These estimates represent the economic cost of all resources required to deliver health services, including staff time, capital and equipment, drugs, supplies, and overhead costs. Quantities of resources used were defined following WHO guidelines and refined based on expert advice to identify less resource–intensive activities in the area of case management that were more feasible in low- and middle-income settings. More information on unit costs calculations can be found in Sections G and H in S1 Text.

Household costs of COVID-19 diagnosis and treatment include out-of-pocket expenses for care seeking, funeral expenses, and productivity losses due to lost income from isolation of cases and were sourced from previously published work [17].

## Outcome evaluation

For our scenarios, we simulate 100 matched replicates sampling from the epidemiological parameter distribution developed by the fitting process. We calculate the resulting epidemiological and economic outcomes (e.g., cumulative DALYs averted, costs and ICERs at annual increments after start of vaccination) for each intervention scenario matched to the corresponding nonintervention scenario (i.e., by draw from the parameter distribution). We then take the relevant quantiles of these simulation outcomes across the samples.

## Results

### Fit to data and epidemic projections without vaccination

Our transmission model is able to fit reported COVID-19 cases and deaths in Sindh for April to September 2020 for different infection-induced immunity assumptions; each gives comparable quality fits (deviance information criterion (DIC) values: no waning protection, 2,771; expected protection 5 years, 2,772; 2.5 years, 2,778; 1 year, 2,766). The model also produces seropositivity comparable to 3 serosurveys in Karachi. At the end of the fitting period, we

estimate 48.1K deaths (95% credible interval (CrI): 45.3 to 49.7K) and 10.5M cases (95% CrI: 9.9 to 10.9M), with ascertainment of 5.3% (95% CrI: 4.8% to 5.8%) of deaths and 1.4% (95% CrI: 1.2% to 2.0%) of cases. Fig 1 shows our baseline assumption of 2.5 years for infection-derived immunity. When the best fitting parameters are used to project cases and deaths beyond September 2020, however, only the shorter durations of protection appear to give a reasonable fit (Fig C in S1 Text).

In forward projections of epidemics between 2022 and 2030 in the absence of vaccination, we found that the duration of immunity following infection is the major determinant of the size of epidemics, as measured by annual incidence (Fig 2). If immunity largely wanes within a year, the region will rapidly settle into recurring epidemics of comparable scale to the 2020 waves. For longer durations of protection, there will tend to be some interannual oscillation. Lifelong immunity results in transmission only at very low residual levels, though we do not consider external reintroductions. This is consistent with epidemic theory, where low immunity duration leads to a rapidly stabilising endemic disease burden, while intermediate durations lead to a series of shrinking epidemic waves settling eventually to lower endemic transmission.

### Impact of vaccination on projected cases and deaths

In our base case scenario, vaccination averts 0.93 (95% CrI: 0.91, 1.0) million cases and 7.3 (95% CrI: 7.2, 7.4) thousand deaths over 10 years (Table 2). We found that the annual cases

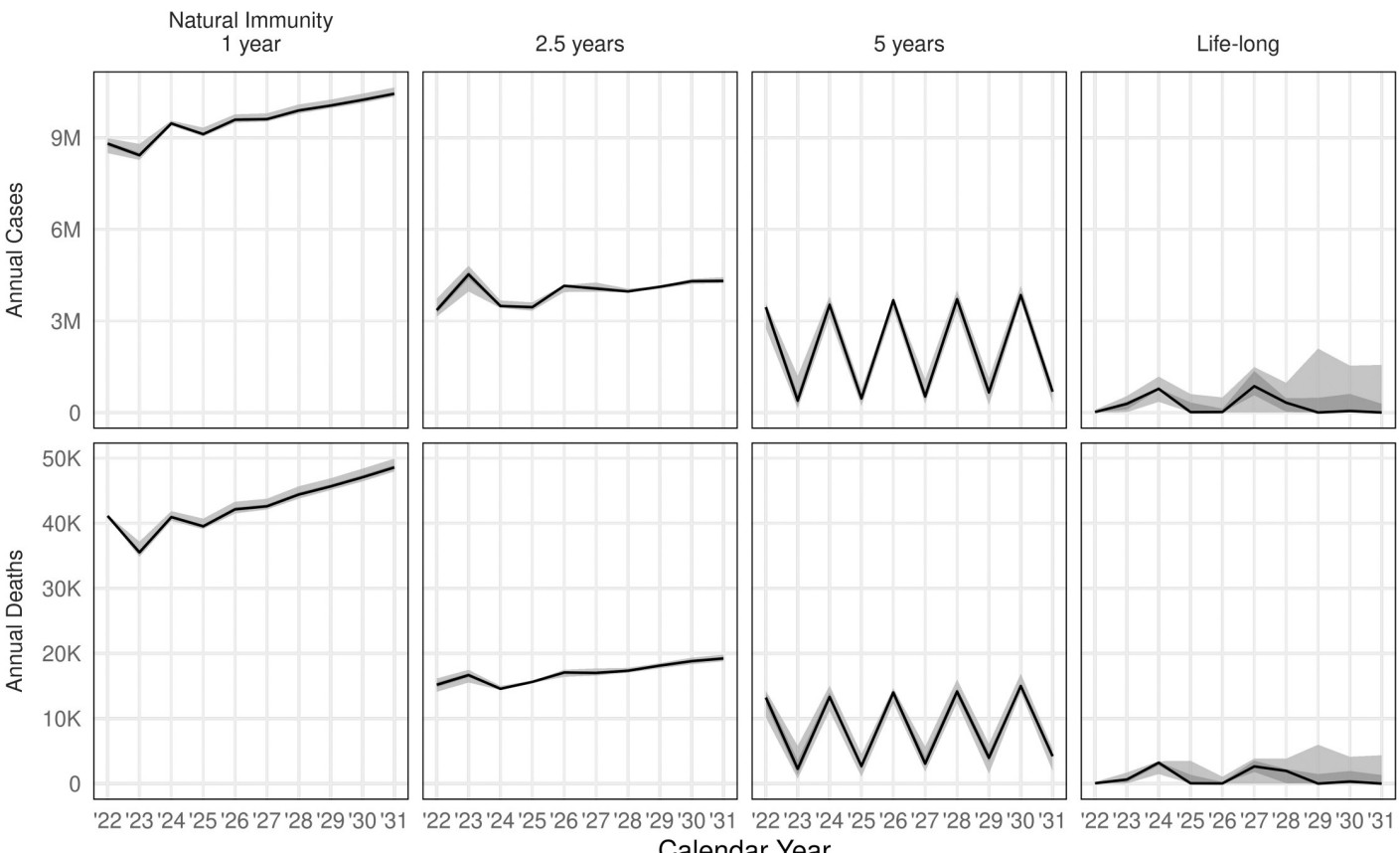

**Fig 2. Long-term baseline projections without vaccination for different assumptions about the duration of natural immunity.** Black line shows median simulation, and grey windows mark 50 and 95% simulation intervals.

**Table 2. Costs in US dollars ($), DALYs averted, cost-effectiveness ratio, cases averted, and deaths averted for different vaccination programme scenarios compared to a counterfactual scenario without vaccination.**

| No. | Description | Impact compared to no vaccination | | | | |
|---|---|---|---|---|---|---|
| | | Cases Averted (millions) | Deaths Averted (thousands) | Difference in Cost ($ millions) | DALYs Averted (thousands) | Cost per DALY Averted ($) |
| **Base case** | | | | | | |
| 1 | Vaccine base case | 0.9 (0.8, 0.9) | 10.1 (10.1, 10.3) | 2.0 (0.1, 2.9) | 70.1 (69.9, 70.6) | 27.9 (1.7, 40.9) |
| **Economic assumptions** | | | | | | |
| 2 | DALYs discounted at 0% | 0.9 (0.8, 0.9) | 10.1 (10.1, 10.3) | 2.0 (0.1, 2.9) | 97.0 (96.8, 97.3) | 20.1 (1.3, 29.5) |
| 3 | DALYs based on higher comorbidities | | | | 54.9 (54.8, 55.4) | 35.5 (2.2, 52.2) |
| 4 | Societal perspective | | | −20.2 (−22.5, −19.1) | 70.1 (69.9, 70.6) | cs (cs, cs) |
| 5 | $6 price per dose | | | 54.7 (52.8, 55.6) | | 780.5 (749.0, 793.0) |
| 6 | $10 price per dose | | | 124.8 (123.0, 125.8) | | 1,781.9 (1,744.2, 1,795.7) |
| **Vaccine and immunity assumptions** | | | | | | |
| 7 | Target 15+ from outset | 1.0 (0.9, 1.0) | 6.4 (6.4, 6.5) | 2.7 (−1.1, 4.3) | 55.9 (55.1, 57.9) | 48.4 (cs, 78.0) |
| 8 | 5-year campaign | 6.0 (6.0, 6.0) | 40.7 (40.2, 41.3) | 85.6 (84.5, 87.3) | 344.8 (341.8, 347.7) | 248.1 (243.6, 255.1) |
| 9 | 10-year campaign | 10.9 (10.9, 11.0) | 67.0 (66.2, 67.9) | 270.3 (267.9, 271.8) | 560.7 (556.9, 565.5) | 482.1 (473.8, 488.1) |
| 10 | Slow rollout: 4K courses per day (no scale-up) for 10 years | 0.7 (0.6, 0.7) | 32.5 (31.7, 33.5) | 38.2 (35.8, 40.1) | 154.0 (150.3, 158.5) | 247.9 (225.9, 267.1) |
| 11 | Fast rollout: 184K courses per day (no scale-up) for 6 months | 3.2 (3.0, 3.3) | 20.1 (20.0, 20.2) | 102.8 (97.2, 112.4) | 181.8 (178.4, 183.2) | 565.6 (530.9, 629.5) |
| 12 | 1-year vaccine and natural immunity waning | 0.9 (0.9, 0.9) | 11.9 (11.7, 12.2) | −5.1 (−6.1, −4.7) | 81.9 (81.0, 83.8) | cs (cs, cs) |
| 13 | 5-year vaccine and 2.5-year natural immunity waning | 1.6 (1.6, 1.7) | 18.4 (18.2, 18.8) | −53.8 (−54.5, −53.4) | 126.8 (125.8, 127.9) | cs (cs, cs) |
| 14 | 1-dose regimen (twice rate of people vaccinated) | 1.7 (1.6, 1.7) | 12.9 (12.9, 13.1) | −48.1 (−53.0, −46.7) | 105.8 (104.8, 108.0) | cs (cs, cs) |
| 15 | 30% vaccine efficacy | 0.4 (0.4, 0.4) | 5.0 (5.0, 5.1) | 36.4 (35.8, 36.9) | 33.8 (33.7, 33.9) | 1,080.7 (1,056.8, 1,091.2) |
| 16 | 90% vaccine efficacy | 1.1 (1.0, 1.1) | 12.2 (12.1, 12.4) | −13.4 (−15.8, −12.3) | 85.5 (85.3, 86.2) | cs (cs, cs) |
| 17 | Vaccine protects against disease-only | 0.6 (0.6, 0.6) | 9.1 (8.9, 9.3) | 14.2 (13.9, 14.4) | 60.1 (59.5, 61.1) | 236.8 (227.3, 242.3) |
| 18 | Vaccine protection is leaky | 0.6 (0.6, 0.7) | 8.2 (8.2, 8.3) | 18.0 (15.8, 19.3) | 55.0 (54.6, 56.0) | 327.4 (281.4, 353.4) |

The base case vaccination scenario assumes the following: a 1-year campaign using a 2-dose vaccine regimen with 70% efficacy at a price of $3 per dose; 2.5-year duration of natural and vaccine induced immunity; and costing from a healthcare perspective.

cs, cost saving; DALY, disability-adjusted life year.

averted by vaccination are higher for longer duration of vaccine-induced immunity, in scenarios targeting people aged over 15 or over 65 for vaccination (Fig 3). For 70% efficacious vaccines that generated 1, 2.5, and 5 years of protection, the median cumulative cases averted was negative in 2022, and for vaccines of 1 and 2.5 years of protection, also negative in 2023. Only duration of immunity of 1 and 2.5 years showed negative deaths averted in 2022 when targeting 15+, and duration of 1 year when targeting 65+.

These temporary negative years are an outcome of delaying a wave of infections (without replacing lost immunity ongoing vaccination beyond the initial year in the base scenario), leading to offset epidemic years, which ultimately has some net reduction in cases and deaths, but in the short-term experiences an epidemic when with no intervention the infection-induced immunity would be preventing that epidemic. In general, a vaccine with low duration of immunity delayed and slowed the oscillation of epidemics but does not substantially reduce total burden because rapid waning leads to relatively low effective coverage.

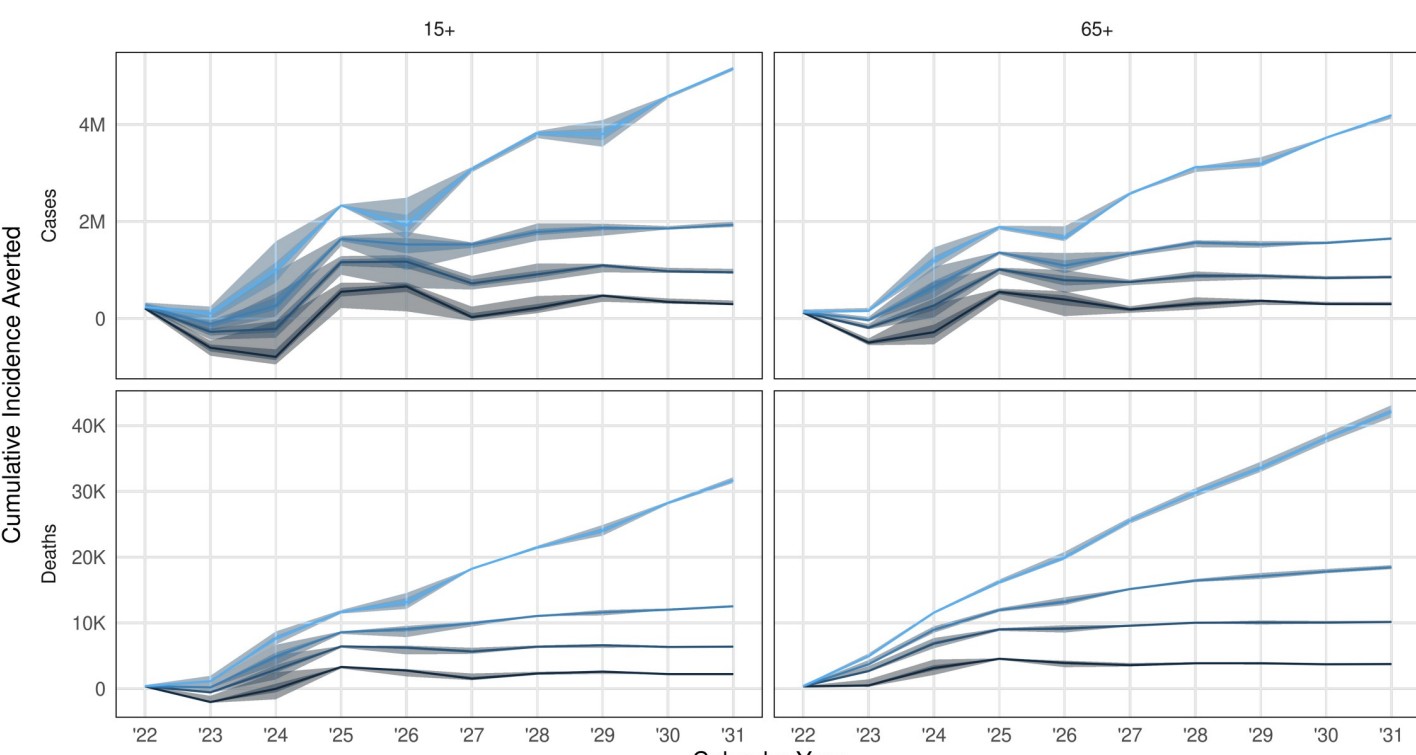

**Fig 3. Cumulative cases and deaths averted by the end of each year.** For a vaccine efficacy of 70%, delivered in a 2-dose schedule over a 1-year vaccine campaign, and expected duration of infection-derived immunity assumed to be 2.5 years, the median averted disease (lines; darker ribbon 50% IQR, lighter ribbon 95% IQR) with varying vaccine protection duration (from dark to light, increasing vaccine protection duration) and initial target age group (either 15+ or 65+; after the first quarter of vaccination, 15+ is targeted in both cases); other scenarios and health outcomes in Figs D and E in S1 Text.

## Economic outcomes of vaccination strategies

The cost of delivery per dose was estimated to be $1.01, excluding vaccine price, wastage, and freight charges. Based on a vaccine price of $3 per dose, the total undiscounted cost of the vaccination programme was estimated to be $64.1 million, $496 million, and $1.04 billion for a 1-year, 5-year, and 10-year campaign, respectively.

The incremental cost, taking into account cost savings from reduced COVID-19 burden, was influenced by the duration of infection-induced immunity (Fig 4). When this duration was short (1 to 2.5 years), then annual incremental costs are likely to be cost-saving in the long run. For longer durations of infection-induced immunity, the duration of the campaign affected the annual incremental costs, with the potential for negative costs at the cessation of 5-year campaigns from a health sector perspective. If the infection-induced immunity is life-long, then the extra protection from the vaccine is of limited benefit. The cumulative number of DALYs averted over the entire 10-year time horizon is positive for all vaccine strategies, although it is especially high for a short duration of natural immunity and long vaccine campaign (Fig 5).

## Cost-effectiveness of vaccination scenarios

Over 10 years, our base case vaccination scenario averts 70.1 (95% CrI: 69.9, 70.6) thousand DALYs, at an additional cost of $2.0 (95% CrI: 0.1, 2.9) million after deducting the cost of the

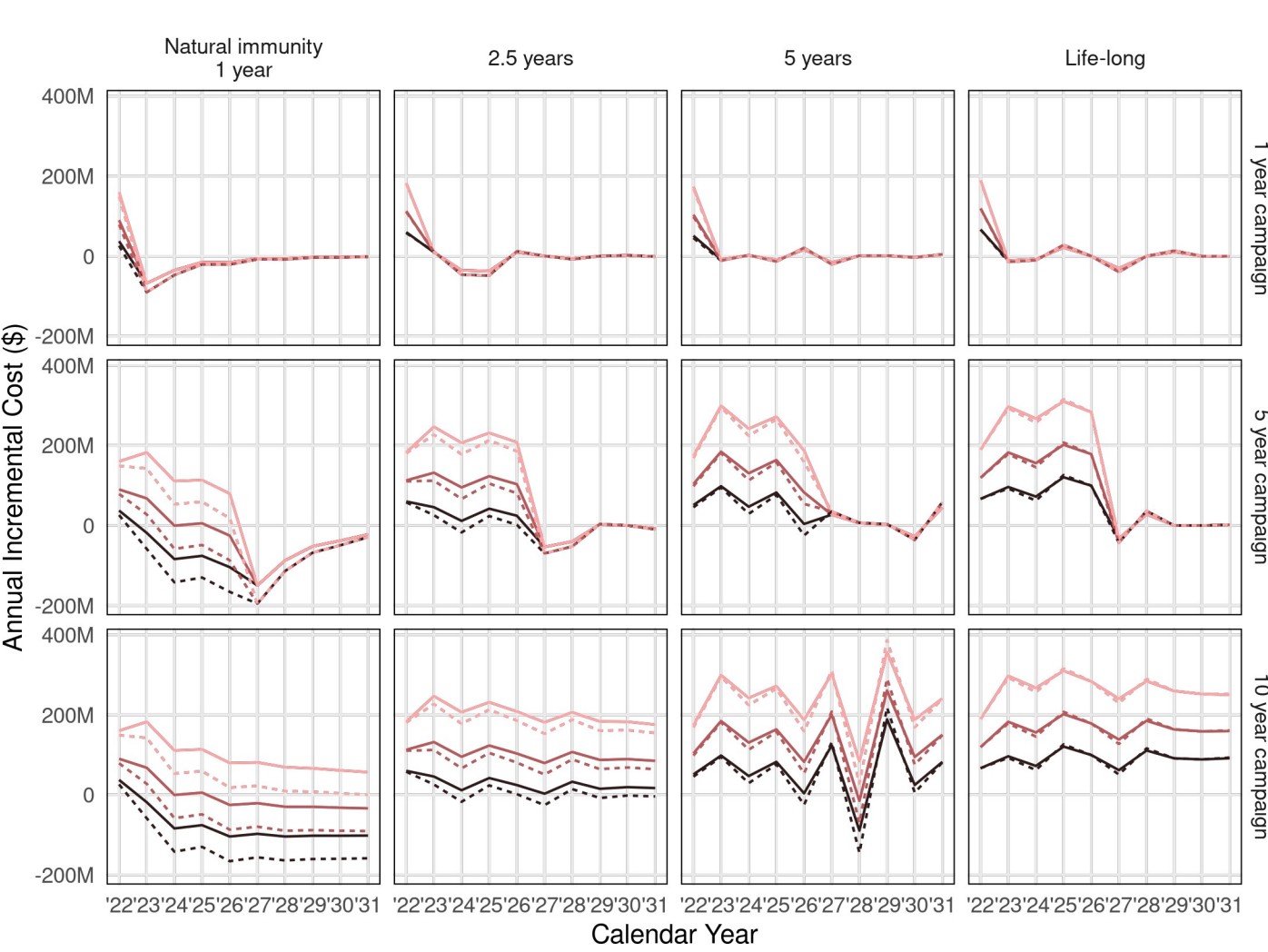

**Fig 4. Annual incremental costs of vaccination programme (compared to no vaccination) for different vaccination strategies and assumptions about the duration of infection-induced immunity.** Results are shown for vaccination using a 2-dose vaccine regimen with 70% efficacy and 2.5-year duration. The societal perspective includes household out-of-pocket payments and lost income but excludes wider economic impacts of the pandemic. Red lines show different vaccine prices, and the solid and dashed lines show health system costs and with societal costs, respectively.

vaccination programme (Table 2). These results are relatively stable when vaccination is not age-targeted (i.e., the entire population 15 years and older are given vaccination from the outset), DALYs are discounted at 0%, or COVID-19 patients are assumed to have a higher rate of comorbidities.

A 1-dose regimen (assuming no loss in efficacy) with twice the rate of people vaccinated results in greater health gains averting 105.8 thousand DALYs and is cost saving. Extending the length of the vaccination campaign to 5 or 10 years also substantially increases health benefits but also leads to higher costs yielding ICERs of $248.1 and $482.1, respectively.

Increasing the vaccine price to $6 or $10 per dose would dramatically increase the net costs, leading, respectively, to ICERs of $781 or $1,782 per DALY averted. On the other hand, a vaccine with higher efficacy, longer duration of vaccine protection, or using a societal perspective would make vaccination cost saving.

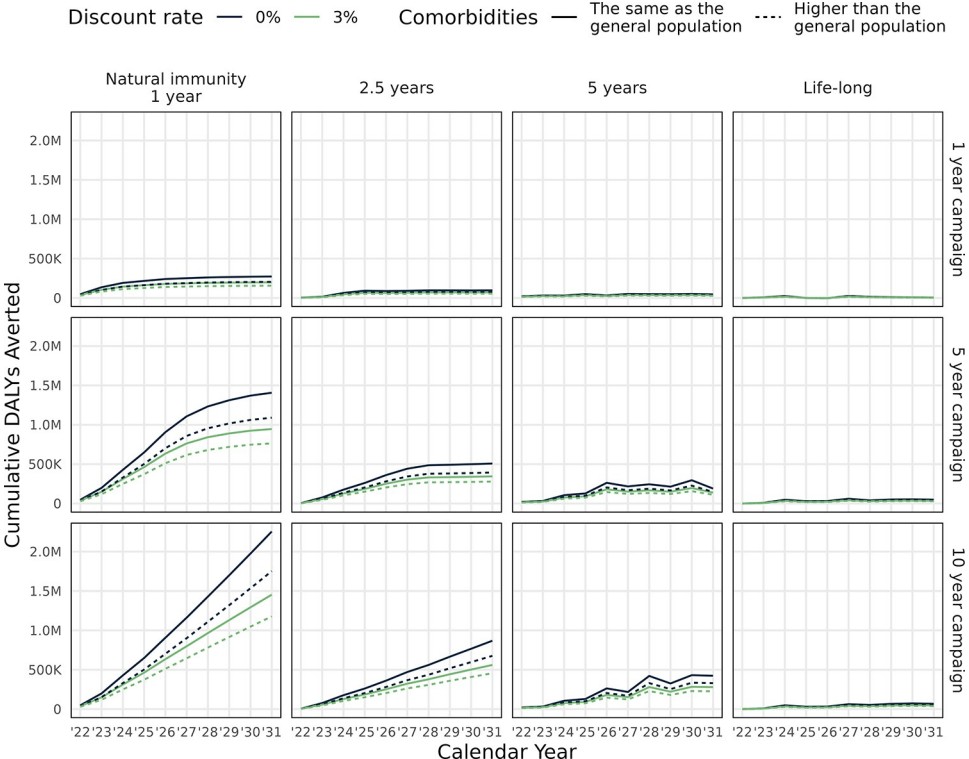

**Fig 5. Cumulative DALYs averted over the 10-year period due to potential vaccination programmes.** For vaccination using a 2-dose vaccine regimen with 70% efficacy and 2.5-year duration. DALY, disability-adjusted life year.

## Discussion

Under a variety of practically relevant epidemiological and economic assumptions, this model of COVID-19 projects that a vaccine programme consistent with the rollout speed projected by COVAX can avert millions of cases and tens of thousands of deaths, and do so in a cost-effective or even cost-saving manner. The particular context considered, Sindh, is a setting with a young population, high SARS-CoV-2 transmission in the past, and limited resources. The age distributions, contact patterns, pandemic history, costs, and income levels of most LMICs are likely more similar to those used in this analysis than to those in comparable studies for high-income countries. While the specific quantities found in this analysis are unlikely to apply explicitly, the qualitative differences from high-income countries likely will.

Reiterating our base case scenario, results: assuming that SARS-CoV-2 does not produce lifelong infection-induced immunity, reaching the COVAX-proposed level of coverage (20%) of an efficacious vaccine (70% efficacy for 2.5 years) within a year (and then ceasing vaccination) may avert 900,000 cases and 10,000 deaths.

Under our base case assumptions, a single year of vaccination would cost an additional $2 million compared to no vaccination and would avoid 70,000 DALYs resulting in an ICER of $28 per DALY averted. This assumes that vaccination can be delivered at $1 a dose, in line with incremental economic cost estimates from the EPIC vaccine delivery costs catalogue, which range between $0.48 and $1.38 for new vaccines [44]. A vaccination campaign extended to 5 years or 10 years would avert substantially more DALYs, but with higher incremental costs, resulting in ICERs of $248 and $482 per DALY averted, respectively. Pakistan does not have a fixed cost-effectiveness threshold. However, a recently conducted exercise defining

Pakistan's Essential Package of Health Services found that over half of the interventions included had an ICER higher than $500 per DALY averted [38,39].

This model also indicates that these benefits are not particularly dependent on the age group targeted. We found that initially prioritising 65+ year olds would avert 60% more COVID-19 deaths compared to vaccinating everyone 15+ years, although the 2 strategies had similar cost-effectiveness since the broader strategy would prevent more cases and more deaths in younger individuals. We did not consider more complex age prioritisation strategies, such as vaccinating 5-year age bands until reaching a particular coverage and then prioritising the next lower band, because such a programme would entail more complex administration costs and need a model validated on age-specific past cases. We do not have the appropriate data to support such an analysis.

The similarity in economic benefits differs from other model-based analyses set in high-income countries [11–13,28], which find that targeting older adults initially would be much more cost-effective. Potential reasons for these differences include the younger age structure of Sindh compared to high-income countries, and the inferred seroprevalence, which was greater than 50% by September 2020. Initial epidemic waves in Sindh (and other settings) may have raised population-level immunity to a point where transmission-reducing vaccination in high-transmission subgroups (i.e., younger, working age) can indirectly protect subgroups at high risk of severe disease (i.e., older, comorbid). While the quantitative benefits decline if the vaccine only provides protection against disease, the conclusions about comparable benefits for either age prioritisation scheme hold.

However, vaccination would be substantially less cost-effective, and potentially not cost-effective from a purely healthcare payer perspective, if the vaccine could only be procured and delivered at $10 a dose or had efficacy as low as 30%. Also, even if a large-scale multiyear mass vaccination programme is cost-effective, it may nonetheless drain scarce financial and human resources from other essential health services. In addition, there are many nonfinancial constraints (e.g., trained personnel), meaning that health opportunity cost may be higher without careful delivery planning. Decisions about vaccination should also take account of other factors besides cost-effectiveness, such as the disproportionately high burden of COVID-19 and related interventions on socioeconomically marginalised groups, and the urgent need to return the economy and society to normal. To effectively inform policy decisions, analyses such as this should be combined with analyses of macroeconomic impact and data on broader societal impacts in a transparent decision framework (e.g., health technology assessment).

In general, our epidemiological projections have relatively narrow uncertainty intervals. While there remains substantial uncertainty on a daily basis, this tends to be offsetting: Cases may shift a little in time, but an annual aggregation results in fairly narrow estimates. These relatively small intervals propagate through the rest of the analysis. These narrow intervals are an accurate reflection of the model assumptions, but the model is fixing many aspects of the real world that are likely to shift unpredictably over the next several years.

As demonstrated by recent emergence of novel variants, the underlying epidemiology may shift, as will technological and social trends, including the relative prices of the inputs to the economic estimation. Variants able to escape vaccine-induced immunity may be introduced either through importation or local mutation. This process is partially addressed by considering loss of infection- and vaccine-derived immunity. For the fastest immunity loss we considered, expected protection durations of a year, a consistently efficacious vaccine (as might be produced by annual updates) can still be cost saving. If variant emergence was more rapid, revaccination with updated formulations might not be able to keep pace, corresponding to lower efficacy. Lower efficacy vaccine (30%) scenarios for rapid protection loss generally resulted in much worse costs per DALY averted (order 1,000s of $ per DALY).

We also assumed that within a particular age group, there is no association between probability of getting vaccinated and risk of disease. This may not be accurate if, e.g., vaccination targets people with comorbidities (and hence higher risk of severe COVID-19 disease), or people who are risk averse (and hence less likely to be infected) are also more likely to get vaccinated. We do not consider future NPIs beyond May 2021 or innovative coordination with vaccination. If there are substantial changes from the impacts integrated into the fitted estimates of local transmission, our projections will not reflect those. Given these core limitations and uncertainties, the intervals ought to be thought of as about our estimate of the central trend conditional on the scenarios, rather than as reflecting the total volatility in the system.

We used a range of scenarios for the duration of natural immunity, although the shortest duration (1 year) best fitted case and death data. This is because the apparent loss of natural immunity may be driven by other factors we did not consider such as behaviour change or emergence of escape variants. If natural immunity is indeed short-lived, this will further strengthen the conclusion that vaccination is likely to be cost saving.

Our findings provide an example of the type of analysis that LMICs can employ to inform vaccination strategies in terms of target populations and financing requirements. While the economic and societal impact of COVID-19 is substantial, the real resource constraints within the health sector in many LMICs mean that vaccination strategies need to balance the current emergency and the longer term needs of the health sector. The slow rate of vaccine distribution is the major impediment to a larger health impact. Administering vaccine doses in line with projected COVAX availability in a province of roughly 50 million people, it would take around 3 years to reach 60% population coverage. Such a long-term programme may not be feasible if vaccine delivery disrupts delivery of other health services, which is a possibility given that the vaccine is targeted at an age group outside the usual Expanded Program on Immunization (EPI). Hence, both short-term rapid response and longer-term consideration about how COVID-19 vaccination can be incorporated in the broader package of essential health services are important in Pakistan and beyond.

## Supporting information

**S1 Text. Supporting information.** Fig A. Model diagram. Fig B. Demographics in the modelled population. Fig C. Alternative infection-induced protection durations. Fig D. Trends in ICU admissions and person-days. Fig E. Trends in general ward admissions and person-days. Table A. Timeline for COVAX dose availability. Table B. Intervention description and unit costs. Table C. Quantities and price of inputs and unit costs per activity. Table D. PPE costs per general ward bed day and per ICU bed day. Table E. Hygiene costs per general ward and ICU bed day. Table F. Pathways of oxygen management. Table G. Disability life years per COVID-19 death. Fig F. Sensitivity of ICER to time horizon under different assumptions about vaccine price, costs, campaign duration, and duration of natural immunity. Fig G. Sensitivity of ICER to time horizon under different assumptions about comorbidities, discounting of DALYs, campaign duration, and duration of natural immunity.
(PDF)

## Acknowledgments

We thank Ulla Griffiths (UNICEF) for information on the cost of COVID-19 vaccine delivery. The authors, on behalf of the Centre for the Mathematical Modelling of Infectious Diseases (CMMID) COVID-19 working group, wish to thank the Defence Science and Technology Laboratory (Dstl) for providing the High Performance Computing facilities and associated

expertise that has enabled these models to be prepared, run and processed in an appropriately rapid and highly efficient manner. Dstl is part of the UK Ministry of Defence.

CHiL COVID-19 Working Group: Mishal Khan, Nichola Naylor, Matthew Quaife, Nuru Saadi, Julia Shen.

CMMID COVID-19 Working Group: Mihaly Koltai, Fiona Yueqian Sun, W John Edmunds, Sophie R Meakin, Samuel Clifford, Kevin van Zandvoort, Yang Liu, Yalda Jafari, Timothy W Russell, Sam Abbott, Graham Medley, Alicia Showering, Katherine E. Atkins, Yung-Wai Desmond Chan, Emily S Nightingale, Anna M Foss, Kiesha Prem, Rachel Lowe, Damien C Tully, Oliver Brady, Rosanna C Barnard, Amy Gimma, Christopher I Jarvis, William Waites, Hamish P Gibbs, Frank G Sandmann, Stefan Flasche, Thibaut Jombart, Joel Hellewell, Jack Williams, Gwenan M Knight, C Julian Villabona-Arenas, Matthew Quaife, Fabienne Krauer, Billy J Quilty, Petra Klepac, Naomi R Waterlow, Sebastian Funk, Akira Endo, Jiayao Lei, Kaja Abbas, Adam J Kucharski, Kathleen O'Reilly, James D Munday, Nikos I Bosse, Alicia Rosello.

## Author Contributions

**Conceptualization:** Carl A. B. Pearson, Fiammetta Bozzani, Simon R. Procter, Maryam Huda, Henning Tarp Jensen, Marcus Keogh-Brown, Muhammad Khalid, Sedona Sweeney, Sergio Torres-Rueda, Rosalind M. Eggo, Anna Vassall, Mark Jit.

**Data curation:** Carl A. B. Pearson, Fiammetta Bozzani, Simon R. Procter, Maryam Huda, Henning Tarp Jensen, Sedona Sweeney, Sergio Torres-Rueda.

**Formal analysis:** Carl A. B. Pearson, Simon R. Procter, Sedona Sweeney, Sergio Torres-Rueda.

**Funding acquisition:** Rosalind M. Eggo, Anna Vassall, Mark Jit.

**Investigation:** Carl A. B. Pearson, Fiammetta Bozzani, Simon R. Procter, Nicholas G. Davies.

**Methodology:** Carl A. B. Pearson, Fiammetta Bozzani, Simon R. Procter, Nicholas G. Davies, Sedona Sweeney, Sergio Torres-Rueda, Rosalind M. Eggo, Anna Vassall, Mark Jit.

**Project administration:** Rosalind M. Eggo, Anna Vassall, Mark Jit.

**Resources:** Rosalind M. Eggo, Anna Vassall, Mark Jit.

**Software:** Carl A. B. Pearson, Simon R. Procter, Nicholas G. Davies.

**Supervision:** Rosalind M. Eggo, Anna Vassall, Mark Jit.

**Validation:** Carl A. B. Pearson, Nicholas G. Davies.

**Visualization:** Carl A. B. Pearson, Simon R. Procter, Nicholas G. Davies.

**Writing – original draft:** Carl A. B. Pearson, Fiammetta Bozzani, Simon R. Procter, Sedona Sweeney, Sergio Torres-Rueda.

**Writing – review & editing:** Carl A. B. Pearson, Fiammetta Bozzani, Simon R. Procter, Maryam Huda, Henning Tarp Jensen, Marcus Keogh-Brown, Muhammad Khalid, Sedona Sweeney, Sergio Torres-Rueda, Rosalind M. Eggo, Anna Vassall, Mark Jit.

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
