## [Editor Report · Decision Letter 0]

24 Feb 2021

Dear Dr Pearson, 

Thank you for submitting your manuscript entitled "Health impact and cost-effectiveness of  COVID-19 vaccination in Sindh Province, Pakistan" for consideration by PLOS Medicine.

Your manuscript has now been evaluated by the PLOS Medicine editorial staff and I am writing to let you know that we would like to send your submission out for external peer review.

Please re-submit your manuscript within two working days, i.e. by February 26, 2021.

Kind regards,

Beryne Odeny

Associate Editor

PLOS Medicine

---

## [Decision Letter · Decision Letter 1]

16 Apr 2021

Dear Dr. Pearson,

Thank you very much for submitting your manuscript "Health impact and cost-effectiveness of  COVID-19 vaccination in Sindh Province, Pakistan" (PMEDICINE-D-21-00936R1) for consideration at PLOS Medicine. 

[LINK]

In light of these reviews, I am afraid that we will not be able to accept the manuscript for publication in the journal in its current form, but we would like to consider a revised version that addresses the reviewers' and editors' comments. Obviously we cannot make any decision about publication until we have seen the revised manuscript and your response, and we plan to seek re-review by one or more of the reviewers. 

We expect to receive your revised manuscript by May 07 2021 11:59PM. Please email us (plosmedicine@plos.org) if you have any questions or concerns.

We look forward to receiving your revised manuscript. 

Sincerely,

Beryne Odeny, 

PLOS Medicine

plosmedicine.org

1. Please revise your title according to PLOS Medicine's style. Your title must be nondeclarative and not a question. It should begin with main concept if possible. Please place the study design (for example, "A cost-effectiveness analysis”, “A modelling study”) in the subtitle (i.e., after a colon).

2. Abstract summary - At this stage, we ask that you reformat your non-technical Author Summary. The Author Summary should immediately follow the Abstract in your revised manuscript. This text is subject to editorial change and should be distinct from the scientific abstract. The summary should be accessible to a wide audience that includes both scientists and non-scientists. Please see our author guidelines for more information: https://journals.plos.org/plosmedicine/s/revising-your-manuscript#loc-author-summary.

3. Abstract:

a. Please structure your abstract using the PLOS Medicine headings (Background, Methods and Findings, Conclusions).

b. Please combine the Methods and Findings sections into one section, “Methods and findings”. Please ensure that all numbers presented in the abstract are present and identical to numbers presented in the main manuscript text.

c. In the abstract, please include the important parameters included in your model.

d. In the last sentence of the Abstract Methods and Findings section, please describe the main limitation(s) of the study's methodology.

4. Please avoid assertions of primacy ("Our study provides the first combined epidemiological and economic analysis ...."). Instead use the phrase, “To our knowledge, this is the first…”

5. Please use the "Vancouver" style for reference formatting, and see our website for other reference guidelines https://journals.plos.org/plosmedicine/s/submission-guidelines#loc-references.

Comments from the reviewers:

Reviewer #1: General Comments

This manuscript contributes to the evidence base on the cost-effectiveness of COVID-19 vaccination relative to no vaccination in low- and middle-income countries. Within a constrained vaccine supply environment, the authors set out to determine the most cost-effective vaccine prioritisation strategy for LMICs. To do this they combined an epidemiological and economic model to assess the health impact, economic impact and cost-effectiveness of Covid-19 vaccination in the Sindh province, Pakistan. The authors show that it is still cost-effective to vaccinate a very small proportion of the population regardless of age-related targeting strategy as long as the vaccine is reasonable priced, efficacy is high, and a reasonable period of natural- and vaccine-induced immunity. This paper provides compelling evidence for the cost-effectiveness of a small-scale COVID-19 vaccination programme irrespective of targeting strategy. 

Minor Essential Revisions

* Vaccine distribution. What does a vaccine campaign of 5 or 10 years mean? Are people vaccinated continuously over the period at a rate of 4000 vaccines/day? Is it assumed that a different segment of the population is vaccinated each year? On page 18, the authors state that "Administering 4000 doses/day in a province of roughly 50 million people would need to be continued for a long time for vaccination to have a large impact". What do the authors mean by 'large impact' - greater reduction in cases and mortality, or closer to herd immunity? At a rate of 4000 doses/day, after 1 year approximately 1.5million (out of a population of 50 million people) will be vaccinated. This is a very small proportion of the population (3%). It would be interesting for the authors to show a scenario that involves a different roll-out from that predicted based on COVAX but results in the ability to vaccinate a higher proportion of the population (that would be closer to a herd immunity target) in one year. Would this still be cost-effective? 

* Rate at which vaccine roll-out occurs. The authors also state that "For simplicity, vaccination occurs at the same rate on every day in the model" and that the "slow rate of vaccine distribution is the major impediment to larger health impact" (page 18). How would changing the rate change the results? How would a slower-rate of roll-out affect the results? 

* Imports: It would appear that the model does not take into account imports - "Do not consider external re-introductions" (page 10). How does this affect the results? This could be a discussion point.

* Variants: How would the inclusion of variants impact the results? This could be added to the discussion - e.g. the effect of variants is likely to reduce the period of acquired immunity - possibly to less than a year? This would make vaccines more cost-effective if they are protective against variants or less cost-effective if it reduces the vaccine immunity to less than a year. Would it be worthwhile examining vaccine protection of less than a year (worst case scenario for vaccination) to determine what protection duration would no longer be cost-effective.

* One dose versus two-dose vaccine. In the sensitivity analysis in Table 2 it would appear that the adoption of a one dose over a two-dose vaccine changes the number of people vaccinated. But how does it impact vaccine immunity - e.g. what if a person receives only the first dose of a two-dose vaccine, or if there is a delay in receiving the second dose? Is the assumption that the second dose is always provided? If so, this needs to be more clearly stated.

* Methods

o The authors have conducted a very detailed in-depth costing of vaccine delivery which is a major strength of the paper. The analysis would benefit from more clearly disaggregating this costing and presenting it succinctly in Table 1 (or in the results) so that it would be possible to see the total cost per dose administered and the different components that make up this total cost. For example, it would be very interesting to see what proportion of the cost per dose the $3 price of the vaccine contributes towards. Currently, it would appear that the major cost item is this vaccine price, as this changes the results of the analysis from cost saving to no longer cost saving.

o Table 1 - what proportion of doses are assumed to be facility-based v. campaign-based? This presumably informs the range for the vaccine cost per dose?

o The authors assume that the vaccine provides protection against infection and disease? How does efficacy play out in the model? Also, the authors use the term "infection-blocking" in the abstract but in the main text describe the vaccine as providing equal protection against infection and disease.

o Table 1 only shows the base case parameter values but a range is used in the analysis (and presented in Table 2). It would be helpful to the reader to list these all here in Table 1. For example, list base case (70%) efficacy as well as low (30%) and high (90%) etc. This should be followed through to the text - for example, authors state "As alternatives, we considered a higher efficacy (90%) …" But, an efficacy of 30% is also evaluated. 

o Organise Table 1 a bit better and maybe include some more of the costs that are mentioned in 'Costs of COVID-19 diagnosis and treatment' section in the table.

o Describe in the methods how the confidence intervals presented in Table 2 are calculated (especially for the ICER).

* Discussion:

o How representative is the Sindh province in Pakistan to other LMICs? How generalizable are these results to other LMICs, for example to LMICs which have lower seroprevalence/higher seroprevalence? 

* Other

o Two full stops at end of second last paragraph on Page 6

o Define NPI bottom of page 7

o Define DIC bottom of page 9. 

o Figure 3 - refers to Figures SY-SZ? Page 13.

Reviewer #2: This was a timely analysis looking at the cost-effectiveness of vaccination in a province in Pakistan. As COVAX begins to distribute vaccines, it's helpful to understand the expected impact and cost-effectiveness of this vaccine distribution. The authors do an excellent job looking at all different types of 

Methods

1. "Leaky protection" - please describe what this means in comparison to non-leaky protection- and the rationale for choosing this approach.

2. "Given the emphasis on prioritising older adults in WHO's vaccine prioritisation roadmap (27), we considered two scenarios for distribution: either individuals 15+ years old for the duration or individuals 65+ years old for the first quarter before shifting to 15+. For all scenarios, we assume vaccine doses are uniformly (i.e., proportional to fraction of population) distributed in the targeted populations." 

I recommend considering a third scenario- which may be particularly in resource limited settings- the impact and cost-effectiveness of targeting those 65+ for the entire duration of the time period.

3. "For COVID-19 deaths we estimated age-specific DALYs using the premature-death method by Briggs (29,30) which builds on standard life-table methods to estimate the discounted years of life lost adjusting for age-related quality-of-life (QoL) in the general population, and also allows for inclusion of different baseline morbidity and mortality assumptions."

Traditional DALY calculations would be difficult in COVID, particularly in LMICs where the average age of COVID death is probably greater than the average life expectancy. I had to read the Briggs paper in depth to be able to understand the approach- I think it's worth a few-sentence summary here and address how this specifically is handled in your application of the Briggs approach. 

4. Scenario analysis: There were many scenarios evaluated, which were helpful. However, what if there is differential immunity for vaccine and natural immunity- like we see with influenza? E.g. what if the duration of vaccine immunity is 2.5 years and duration of natural infection immunity is 10 years? To what degree does this impact cost-effectiveness?

5. Distribution assumption: The fact that this model examines the COVAX distribution is very useful, particular for the early phase of vaccination. However, once high-income countries feel satisfied with their vaccination rates (ugh. so much for vaccine equity.), there may be more vaccine available for LMICs (or could be advocated for). How much more cost-effective would the vaccination campaigns be if you could double/triple/quadruple the number of vaccines per day (say, starting in mid 2022?)? If this is shown to be vastly (or even marginally) more cost-effective, the findings of this manuscript could be used to further advocate for more vaccines through COVAX or other distribution mechanisms. 

I could also imagine a scenario in which you have a massive vaccination campaign every so-many-years (depending on the duration of vaccine-induced immunity)- and that could be substantially more cost-effective than a slow drip of immunization. (So, what if we could vaccinate most of the 65+ population every 5 years or so?) Any of these different distribution scenarios could have the power to change and advocate for vaccines for LMICs and I strongly suggest the authors pursue these scenarios.

Minor comments:

1. "For the non-fatal outcomes, and in the absence of specific DALY data, we used Quality Adjusted Life Years (QALYs) reported by Sandmann et al. (28) based on pandemic influenza studies treated one QALY as equivalent to one DALY averted."

I suggest adding the word 'gained': "… treated one QALY gained as equivalent to one DALY averted."

Reviewer #3: This study estimated COVID-19 cases and death over 10 years under various vaccine scenarios in a population of 48 million people for a Pakistani province. The simulation model is a previously published compartmental transmission model under an extended SEIRS+V structure including birth, death, and age-strata. The model is calibrated to new daily cases and deaths in the Sindh province from April to September 2020 and validated until January 2021. The authors conducted a cost-effectiveness analysis of various vaccine scenarios compared to no vaccination over 10 years. Sensitivity analyses include the length of vaccination campaign, cost per dose, natural immunity loss and duration of vaccine protection, etc. The study concluded that COVID-19 vaccination is likely to be cost-effective if the cost is low and vaccine has good protection against infection in low- and middle- income countries. 

This study is well done. Long-term model projections under various natural immunity loss and vaccine protection waning scenarios are particularly insightful. I have the following comments to the authors.

Major comments:

1. Contact patterns changes were estimated using Google Community Mobility indicators and school closures were considered using government response tracker. The authors assumed there is no further social distancing measures after May 2021. This is a very strong assumption. Have the authors considered face mask use data in Sindh and its effect on transmissibility? Given recent government interventions responding to third waves of infection in Europe and face mask use recommendations in many countries, I recommend the authors at least adding a discussion on continuous use of non-pharmaceutical interventions and prolonged changes in contact patterns after 2021.

2. There are several optimistic assumptions about vaccine. First, 10% wastage was assumed. I would like to see if there is any evidence backing up this assumption which seems to be very low. Second, the authors assumed that an effective vaccine provides full protection against infection. I would recommend an additional scenario analysis that relaxes the full protection assumption or a more in-depth discussion on this point. For example, if the vaccine is disease-modifying only, or gives partial protection against infection and reduces transmissibility. 

3. One page 6, could the authors clarify this vaccination prioritization strategy, "individuals 65+ years old for the first quarter before shifting to 15+"? Does this mean 65+ only in the first quarter of the first year of vaccination, or prioritized annually like a seasonal flu vaccination scenario?

4. In the base case analysis, future cost is discounted at 3% annually, and health outcomes are discounted at either 0% (base case) or 3%. Equal rate of discounting is the more common practice, though there are some debates about this practice (https://www.ncbi.nlm.nih.gov/pmc/articles/PMC5999124/). Was there a particular reason that the authors decided to not discount health outcomes in the base case?

5. The author mentioned in the Discussion that 4000 doses/day would need to be continued for a long time to have a large impact. It would be more informative to give readers some ideas on the vaccine coverage over time. 

Reviewer #4: This is an interesting, well-conducted, and timely analysis on the impact of targeted vs general COVID vaccination strategies in low and middle-income countries.

My major comment is that the authors evaluated only two scenarios of vaccine distribution: 1) >65 first followed by the entire adult population, and 2) all persons 15+. Given the complex tiers of vaccine rollout in high-income settings (stratified by healthcare worker status, age, age + comorbidities, essential workers, etc) it would be useful to project the impact of other distribution strategies. For example, the authors find that vaccination of individuals >65 prevents slightly more deaths than mass vaccination, which is not surprising given the smaller proportion of older individuals in low-income settings. However, would vaccinating individuals >50 provide greater benefits? What would be the added benefit of a tiered approach with respect to age and co-morbidities? Projections of vaccine benefit by population heterogeneity (particularly co-morbidities) may be difficult due to the simpler compartmental model design, but it would be helpful to discuss the potential impacts of these strategies or their benefit over a mass vaccination approach in the discussion section.

It would be helpful for the authors to address the impact of vaccine hesitancy and the potential correlation between risk-taking behavior and refusal to uptake vaccination. In particular, vaccinations have a complex political history (ie with polio eradication efforts) and targeted violence toward vaccination workers. https://www.npr.org/sections/goatsandsoda/2021/02/24/968730972/pakistans-polio-playbook-has-lessons-for-its-covid-19-vaccine-rollout

https://gallup.com.pk/wp/wp-content/uploads/2021/01/Gallup-Covid-Opinion-Tracker-Wave-9-pdf.pdf

Given a high distrust of COVID-19 vaccination, would a country such as Pakistan avert greater illness using mass vaccination for all age groups? Perhaps the impact of vaccine refusal rates can be explored in a sensitivity analysis. Perhaps this can also be mentioned in the discussion section.

The authors assume an exponential distribution for waning immunity—this would imply a fast waning early on. It is possible that immunity declines more slowly at first. What are the implications of choosing this distribution?

Introduction:

The authors state: "For all vaccine scenarios, we assumed the vaccine provides protection against infection (not just disease) and that protection is tested with each exposure in the model (i.e. "leaky" protection).

It would be helpful to define the concept of leaky vaccine more clearly for a lay audience of policymakers and researchers.

Do the authors have data on the age distribution of comorbidities associated with COVID-19 severity in Pakistan? If so, how were these data incorporated into the model?

The population mixing assumptions would likely have a strong impact on the results, particularly for herd immunity. Can the matrix assumptions be varied? If not, what are the likely implications of misspecifying the matrix, in terms of choice of vaccination strategy?

For Table 1, it would be useful to list the range of values used in the sensitivity analyses in addition to the base case values.

Much of the discussion section provides a summary of the findings. It would be useful to also contextualize the finding in terms of the strengths and limitations of the model and inputs used. How generalizable are the findings to other countries and what are the factors that most impact generalizability? Eg age structure or mixing in the population?

Reviewer #5: This study aims to assess the health impact, economic impact, and cost-effectiveness of COVID-19 vaccination in Sindh province, Pakistan, using a combined epidemiological and economic model. 

Comments:

The authors apply a previously published compartmental model, providing the relevant citation and a concise summary in brief here.

A technically appropriate methodology of Bayesian inference via Markov Chain Monte Carlo has been used to fit elements of the model, and the authors have undertaken a thorough set of out-of-sample validations.

"Vaccine doses are distributed amongst individuals in the Susceptible and Recovered compartments; Susceptible individuals become Vaccinated, while Recovered are unchanged."

Can the authors please explore and discuss whether, by leaving Recovered unchanged, it is realistic to assume that the Recovered population is the same as the Recovered and Vaccinated population within the model? 

The authors appropriately acknowledge the limitation of the current model by not including the possible impact of variants.

"As demonstrated by recent emergence of novel variants, the underlying epidemiology may shift, as will technological and social trends, including the relative prices of the inputs to the economic estimation. Given that core uncertainty, the intervals ought to be thought of as on our estimate of the central trend, rather than as reflecting the volatility in the system."

The authors have suitably provided the CHEERS checklist.

[LINK]

---

## [Decision Letter · Decision Letter 2]

22 Jun 2021

Dear Dr. Pearson,

Thank you very much for submitting your manuscript "COVID-19 vaccination in Sindh Province, Pakistan: a modelling study of health impact and cost-effectiveness" (PMEDICINE-D-21-00936R2) for consideration at PLOS Medicine. 

[LINK]

In light of these reviews, I am afraid that we will not be able to accept the manuscript for publication in the journal in its current form, but we would like to consider a revised version that addresses the reviewers' and editors' comments. Obviously we cannot make any decision about publication until we have seen the revised manuscript and your response, and we plan to seek re-review by one or more of the reviewers. 

We expect to receive your revised manuscript by Jul 13 2021 11:59PM. Please email us (plosmedicine@plos.org) if you have any questions or concerns.

We look forward to receiving your revised manuscript. 

Sincerely,

Beryne Odeny, 

PLOS Medicine

plosmedicine.org

Thank you for addressing editorial requests.

Comments from the reviewers:

Reviewer #2: 

The authors have substantially improved the manuscript in response to reviewer comments. However, upon re-read of the updated version, in light of everything we know about vaccination at this moment in time I have one major concern:

Who is asking whether or not vaccination is cost-effective (and is basing vaccine delivery strategies based on cost-effectiveness)? Across all LMICs that I've worked with throughout the pandemic, the cost of the vaccine does not seem to be the limiting factor in all of this (particularly as the cost/dose is not the $10 that the authors cite as a worst-case scenario) [https://www.who.int/docs/default-source/coronaviruse/act-accelerator/covax/costs-of-covid-19-vaccine-delivery-in-92amc_08.02.21.pdf]. The costs are low, the effect is clearly high. The real issue remains access to vaccines in the first place (while high income countries eat up the world supply) and within country distribution.

I do see some marginal utility in this analysis with regard to type of scale-up strategy (65+ versus everyone)- and the analyses were all very elegant… but ultimately, by the time more guidance is needed, particularly with the introduction of variants that may reduce the response to current vaccines (as the authors have added into the discussion)- the question on policy maker's minds may then rather (and rightly) become the cost-effectiveness of booster vaccines for some proportion of the population instead of just focusing on initial vaccines. 

The authors should do a more robust job convincing the reader why this one regional analysis is important, and what it adds to the literature - or adapt the model explicitly to look to the question of variants and boosters.

Minor revisions:

Methods: Suggest changing "highly developed settings" to "high income countries"

Reviewer #3: The authors have addressed all my comments from the last round. 

Reviewer #4: The authors have sufficiently addressed the reviewer comments in this revised manuscript. I have a few additional suggestions. 

Abstract:

"Varying these assumptions, we generally find that prioritizing the older (65+) population

prevents more deaths, but broad distribution from the outset is economically comparable in

many scenarios, and either scheme can be cost-effective for low per-dose costs."

This sentence is a bit long and hard to follow.

"However, high vaccine prices ($10/dose) may not be cost-effective." 

This statement seems very tentative. Can it be made more definitive? For example: We found vaccination was not cost-effective at high vaccine costs (10/dose). 

"These projections are limited by the mechanisms present in the model."

This sentence is a bit vague and can be removed. The following sentence could start with: Limitations of this study include…

"Preventing severe disease is an important contributor to this impact, but the advantage of focusing initially on older, high-risk populations may be smaller in generally younger populations where many people have already been infected, typical of many low- and -middle income countries, as long as vaccination gives good protection against infection as well as disease." 

This is a very long sentence. It would be helpful to divide it into a few shorter sentences and clarify the conclusion in more certain terms.

"What Do These Findings Mean?"

This section can be modified to incorporate the implications of the study on COVID vaccination policy.

It might be worth mentioning in the discussion section that other age stratified vaccination scenarios were not modeled because of the lack of epidemiologic data and the difficulty of conducting stratified vaccination for most health systems. Otherwise it may be somewhat intitutive that the health impact of prioritizing persons >65 years in LMICs is low because of the small number of people in that category.

Methods, costs

"Following WHO guidelines, we used a 3% discount rate for future costs and for annualising

capital investments, while health outcomes are discounted at either 0% (base case) or 3%

(35)." 

Most economic guidelines recommend using the same discount rate for health benefits and costs. It would be useful for the authors to provide an explanation behind the rationale to not discount health benefits (but only discount costs) in their sensitivity analysis. It would be useful to provide a sensitivity analysis where neither costs nor health benefits are discounted.

Table 1

It would be helpful for the authors to conduct more sensitivity analyses as many of the parameters are highly uncertain, particularly COVID natural history, vaccination efficacy/duration. These could be presented in the supplemental appendix and referenced in the text. Costs could also be varied more widely. Vaccination is cost-effective at $3 per dose but not cost effective at $10 per dose—this is a wide range. The ICER for vaccination at $10 per dose is very high. At what approximate threshold does the vaccine no longer become cost effective? Similarly, a 30% vaccine efficacy is unlikely but what is the impact of a 50% efficacious vaccine?

Table 2:

It may be helpful to replace "dom" with "cs" for cost-saving as dom usually refers to dominated (a strategy that is more costly and less cost-effective).

Discussion:

"Many lower- and middle-income settings have similar age distributions and

contact patterns, pandemic history, costs, and income levels. As such, we expect these

qualitative conclusions to apply broadly, though with detailed quantitative outcomes

depending on the location-specific values for those parameters."

Could the authors be more specific about which countries these results would be generalizable to? Other countries in South Asia, or Asia? Or Asia and Africa?

"Under our base case assumptions, a single year of vaccination would cost an additional

USD 2 million compared to no vaccination, and would avoid 70,000 DALYs resulting in an

ICER of USD 28 per DALY averted."

Could these results be specified in cases and deaths averted since DALYs are harder to understand intuitively?

"This model also indicates that these benefits are not particularly dependent on the target

population."

I don't think this can be concluded without conducting an age stratified analysis and including healthcare workers, which is not done in this present study. It is unlikely that vaccinating the general population would result in higher benefits than vaccinating healthcare workers.

Reviewer #5: The authors have satisfactorily responded to each comment in turn, and appropriately adjusted the model to include an Recovered-and-vaccinated (RV) compartment.

[LINK]

---

## [Decision Letter · Decision Letter 3]

5 Aug 2021

Dear Dr. Pearson,

Thank you very much for re-submitting your manuscript "COVID-19 vaccination in Sindh Province, Pakistan: a modelling study of health impact and cost-effectiveness" (PMEDICINE-D-21-00936R3) for review by PLOS Medicine.

I have discussed the paper with my colleagues and the academic editor and it was also seen again by three reviewers. I am pleased to say that provided the remaining editorial and production issues are dealt with we are planning to accept the paper for publication in the journal.

[LINK]

We look forward to receiving the revised manuscript by Aug 12 2021 11:59PM.   

Sincerely,

Beryne Odeny, 

Associate Editor 

PLOS Medicine

plosmedicine.org

Requests from Editors:

1. Please format your references as follows:

a. Please use the PLOS Medicine style reference call outs throughout the text, noting the absence of spaces within the square brackets, e.g., "... every country) [1,2]." https://journals.plos.org/plosmedicine/s/submission-guidelines#loc-references.

b. Please ensure that journal name abbreviations match those found in the National Center for Biotechnology Information (NCBI) databases, and are appropriately formatted and capitalized. https://journals.plos.org/plosmedicine/s/submission-guidelines#loc-references. Please ensure that weblinks are current and accessible to date.

2. Please consistently define abbreviations in tables and figures e.g., USD, DALYS

Comments from Reviewers:

Reviewer #1: The authors have addressed all my comments from the last round. I however feel that the authors still need to adjust table 1 to reflect the "15% wastage for vaccines, and the 10% wastage for immunization supplies" as indicated in the text.

Reviewer #4: The authors have sufficiently addressed the reviewer comments in this revised manuscript.

[LINK]

---

## [Editor Report · Decision Letter 4]

14 Sep 2021

Dear Dr Pearson, 

On behalf of my colleagues and the Academic Editor, Dr. Brooke E Nichols, I am pleased to inform you that we have agreed to publish your manuscript "COVID-19 vaccination in Sindh Province, Pakistan: a modelling study of health impact and cost-effectiveness" (PMEDICINE-D-21-00936R4) in PLOS Medicine.

PRESS

Sincerely, 

Beryne Odeny 

Associate Editor 

PLOS Medicine